# The Status of Occupational Stress and Its Influence the Quality of Life of Copper-Nickel Miners in Xinjiang, China

**DOI:** 10.3390/ijerph16030353

**Published:** 2019-01-27

**Authors:** Yuhua Li, Xuemei Sun, Hua Ge, Jiwen Liu, Lizhang Chen

**Affiliations:** 1XiangYa School of Public Health, Central South University, Changsha 410000, China; lyh740322@sina.com; 2College of Public Health, Xinjiang Medical University, Urumqi 830011, China; sunsun9010@163.com (X.S.); gehua2710@sina.com (H.G.); liujiwendr@163.com (J.L.)

**Keywords:** copper-nickel miners, occupational stress, quality of life

## Abstract

The purpose of this study was to investigate the status of occupational stress and its influence on the quality of life of copper-nickel miners, in order to provide a theoretical basis for alleviating occupational stress to improve their quality of life. Stratified cluster sampling and a self-administered questionnaire survey were used. The Effort–Reward Imbalance (ERI) questionnaire and the SF-36 (36-Item Short Form) health survey scale were administered to all 2000 miners registered with a copper-nickel mining human resources department and who had been on duty for more than one year. In total, 1857 valid questionnaires were collected, with a response rate of 92.85%. The percentage of the copper-nickel miners suffering from occupational stress was 42.65%. A statistically significant difference was observed in relation to the prevalence of occupational stress among miners of different genders, ages, education levels, and operating units. The occupational stress detection rate was higher for males than females. Miners aged between 30 and 34 years exhibited the highest level of occupational stress compared to other age groups. Those with a junior college education exhibited the highest rate of occupational stress compared to those with other levels of education. Those working in the smelting unit exhibited the highest rate of occupational stress compared to those working in other operational units. Those classified as experiencing stress (an ERI score >1) had lower quality of life scores than miners classified as not experiencing stress (an ERI score ≤1). The results show that level of education, monthly income, and degree of occupational stress affect quality of life among copper-nickel miners. It was found that older age, lower income, higher education level, and higher degree of occupational stress were factors related to poorer quality of life. Copper-nickel miners have high levels of occupational stress, and occupational stress is a risk factor that can diminish quality of life.

## 1. Introduction

Occupational stress refers to the physiological and psychological pressure caused by the imbalance between objective needs and an individual’s capacity to adapt under certain occupational conditions [1]. With the development of society, the pace of modern life is gradually accelerating, and the mode of work is also changing. People are experiencing pressure in relation to family, work, education, health, and other areas, leading to occupational stress among the employed population. As one of the typical harmful factors identified by professional psychology, occupational stress has become a significant problem that is the focus of current international occupational health and psychology research and occupational disease legal planning [2]. The World Health Organization considers occupational stress a worldwide epidemic [3].

There are many factors that contribute to occupational stress. Such factors are mainly classified as occupational or individual. First, occupational factors include: (1) working conditions, including shifts or irregular working hours, sedentary, repetitive, or monotonous working methods, individual or interactive tasks, and controllable or uncontrollable tasks, etc.; (2) working environment, including physical conditions such as heat, noise, and lighting, and chemical conditions such as odor; (3) interpersonal relationships at work, relationships with colleagues or superiors; (4) role in organization, including role ambiguity, role conflict, role overload, and identity consistency; and (5) career development, including job satisfaction, possibility of reward and promotion, job security, certainty of future work, etc. [4,5,6]. Second, individual factors include: individual characteristics (such as gender, age, type A personality), self-perception, self-confidence, and ability to cope with stress [7,8].

Research shows that occupational stress has a dual effect. Moderate occupational stress is conducive to stimulating people’s enthusiasm for work, improving their work efficiency, and ensuring that workers maintain a good psychological and physiological state [9]. However, long-term non-moderate occupational stress is harmful to physical and mental health as well as to the quality of life of professional workers, which can lead to a series of abnormal physiological, psychological, and behavioral reactions among workers [10]. The main symptoms of physiological diseases are hypertension, ischemic heart disease, impairment of an individual’s immune function, impairment of the nervous system and digestive system, etc. [11,12,13,14,15,16,17]. Psychological symptoms primarily include depression, anxiety, burnout, irritability, lack of concentration and so on, which reduce the coping ability of employees [18,19,20]. Behavioral abnormalities manifest in the form of both individual- and organizational-related aspects. Individuals may avoid work, abuse alcohol or drugs, experience a loss of appetite, and engage in hostile behavior; organizational performance is marked by absenteeism, accidents, and a low labor capacity [21].

The loss caused by occupational stress is enormous. It can harm not only the physical and mental health and quality of life of workers, but can also result in financial losses for enterprises and society. Studies have shown that the annual cost of treating diseases caused by occupational stress in the United States is US$500–1000 billion. The United Kingdom loses millions of working days each year due to occupational stress disorder [22,23,24,25]. The International Labor Organization (ILO) estimates that the per annum economic losses caused by occupational stress amount to approximately US$300 billion. In addition, studies have shown that occupational stress is an important reason for staff absence [26]. The impact of occupational stress on occupational groups has become a serious global problem. 

Quality of life reflects the overall feeling related to an individual’s ability to live and work, and to the health of society. Studies have shown that long-term non-moderate occupational stress can harm physical and mental health, adversely affecting the working ability and quality of life of the occupational population [27,28]. Zhang Ying’s research examined occupational stress and found that higher levels of stress among medical staff led to poorer quality of life, and occupational stress is an important factor affecting the quality of life of medical staff [29]. A study carried out by Malamardi showed that occupational stress is associated with quality of life in the banking sector, and that interventions are required to mitigate the occupational stress experienced by employees [30].

Miners belong to a special occupational group whose place of employment is relatively remote. The working environment is relatively poor, labor intensity is high, working hours are long, and the social status of such workers is low. The working environment can easily contribute to varying degrees of occupational stress that affect the quality of life of employees [31]. At present, scholars have conducted extensive studies on occupational stress, although relatively little research has examined the occupational environment of copper-nickel miners. This study administered a questionnaire survey to copper-nickel miners in Xinjiang, China, to investigate the status of occupational stress and its influence on quality of life, to provide a theoretical basis for alleviating occupational stress and improve the quality of life of copper-nickel miners.

## 2. Materials and Methods

### 2.1. Participants

This cross-sectional study was carried out from June 2016 to September 2017. Stratified cluster sampling was employed in accordance with the main production process areas of the copper-nickel mine: the mining unit, ore dressing unit and smelting unit. A self-administered questionnaire was used in conjunction with all workers registered by the human resources department of a copper-nickel mine in Hami City, Xinjiang Uygur Autonomous Region, China, all of whom had been on duty for more than one year. A total of 2000 questionnaires were distributed in this survey. According to the inclusion and exclusion criteria, 1857 valid questionnaires were collected. The response rate of the questionnaires was 92.85%. Inclusion criteria specified no psychiatric or hereditary diseases. Incomplete questionnaires were excluded. The respondents volunteered to participate in the survey and provided their written informed consent. There are seven types of work in this survey, but the time and dose of exposure to occupational hazards in the safety department, infrastructure department, institutional department, and logistics department are much lower than in the mining unit, ore dressing unit, and smelting unit, so only three main types of work were selected for comparison and analysis.

### 2.2. Research Methods

A questionnaire (detailed below) was used to investigate the status of occupational stress and its impact on quality of life.

#### 2.2.1. General Investigation

This section included general demographic characteristics, such as age, sex, marital status, educational level, income, and type of work.

#### 2.2.2. Occupational Stress Investigation

The self-administered Chinese version of the Effort–Reward Imbalance (ERI) questionnaire was adapted from the ERI model, which was developed by Siegrist J [32,33]. The theoretical basis of this model is whether the individual’s work effort is equal to the reward (reward, respect, job prospects and stability) he receives from his work. When the effort is too high and the reward is too low, the individual is considered to be in a tense state. On the contrary, when the effort is too low and the reward is too high, there is no occupational stress. The questionnaire consisted of three sections: effort (E, six items), reward (R, 11 items), and over-commitment (six items), totaling 23 items. A Likert grade 5 scoring method was used, such that a score of 1 indicated “highly disagree” and 5 indicated “highly agree”. In the effort–reward evaluation method, each item is attributed the same weight, and the index of the effort–reward ratio is ERI = E/R × C, where C is the adjustment coefficient (the ratio of the number of items of molecule to the number of items of denominator). In this paper, C = 6/11. If the ERI > 1, this indicates a high effort–low reward return. If the ERI = 1, this indicates a balanced effort–reward return. If the ERI < l, this indicates a low effort–high reward return. The higher the ratio of the ERI, the higher the level of occupational stress. An ERI score that is greater than 1 suggests occupational stress [34,35].

#### 2.2.3. Quality of Life Investigation

The quality of life of copper-nickel miners was evaluated using the SF-36 scale, which is based on the Medical Outcome Study Short Form (MOS-SF) developed by Stewart in 1988 and the Boston Health Research Center in the United States. It is regarded as a measurement tool by which to evaluate quality of life, and has a wide range of applications. The SF-36 scale evaluates eight aspects of Health-Related Quality of Life (HRQOL), i.e., physiological functioning (PF), role–physical functioning (RP), bodily pain (BP), general health (GH), vitality (VT), social functioning (SF), role–emotional functioning (RE), and mental health (MH). Scoring method: According to the different weights attributed to each item, the SF-36 scoring method calculates the sum of the integrals of each item in each dimension, obtains the integrals of eight dimensions, and then converts the integrals into the final score ranging from 0 to 100. High scores in each dimension and high overall scores indicate a better quality of life [36].

### 2.3. Quality Control

In China, workers are required to undergo professional health examinations on a regular basis under the Industrial Safety and Health Act. During the annual professional health examination, we conducted face-to-face interviews with each participant to fill in the questionnaires. Before the formal investigation, we contacted the organization under investigation in order to foster cooperation and facilitate a small-scale pre-investigation. Investigators explained the purpose, significance, content, and requirements of the study to the respondents to obtain their cooperation. Questionnaires were completed and collected on-site by the investigators. They were then promptly checked, and incomplete questionnaires were excluded.

### 2.4. Statistical Methods

Statistical analysis was performed using SPSS 21.0 (SPSS Inc., Chicago, IL, USA). Mann–Whitney U test was used to compare two groups and Kruskal–Wallis H test was used to compare more groups of nonparametric (non-normally distributed) data. A chi-squared test was used for the counting data, and multiple linear regression was employed for multivariate analysis. The significance level (α) was set at 0.05.

## 3. Results

### 3.1. General Demographic Characteristics of Copper-Nickel Miners

Among the 1857 copper-nickel miners, 1635 were men (88.05%) and 222 were women (11.95%). The average age was 33.02 ± 9.52 years (Table 1).

### 3.2. Comparison of Occupational Stress Levels in Different Populations

Occupational stress is considered present when ERI > 1. The survey results show that 42.65% of the miners experience occupational stress. The detection rate of occupational stress was higher for males than females (*p* = 0.004). The detection rate of occupational stress between the ages of 30–35 years was higher than that observed among other age groups (*p* = 0.043). The detection rate of occupational stress with a junior college education was higher than that of the other groups (*p* < 0.001) (Table 2).

### 3.3. Comparison of Occupational Stress among Copper and Nickel Miners Working in Mining, Ore Dressing, and Smelting Units

The rate of occupational stress was highest among those miners working in smelting units (*p* = 0.049) (Table 3).

### 3.4. Comparison of the Quality of Life of Copper and Nickel Miners Working in the Mining, Ore Dressing, and Smelting Units

There were statistically significant differences in the quality of life of copper-nickel miners working within the three different units. Copper-nickel miners working in mining units scored the highest in PF, RP, BP, and RE, while copper-nickel miners from ore dressing units had the highest scores in GH, VT, SF, and MH (*p* < 0.001). The results suggest that copper-nickel miners working in mining units enjoy better physiological health, and miners working in ore dressing units have better mental health (See Table 4).

### 3.5. Comparison of the Quality of Life among Stressed and Non-Stressed Copper-Nickel Miners According to their ERI Score

Stressed (an ERI score >1) miners had lower quality of life scores compared to non-stressed (an ERI score ≤1) miners, suggesting that stress could reduce the quality of life of workers (*p* < 0.001) (Table 5).

### 3.6. Exploration of Factors Influencing Quality of Life

Multiple linear regression was used to analyze the effects of different characteristics of the sample population as well as the way in which the occupational stress of copper-nickel miners influence quality of life. The results show that age, level of education, income, and ERI affect quality of life among copper-nickel miners (*p* < 0.001). Older age, lower income, higher education level, and higher ERI are factors related to poorer quality of life (See Table 6 and Table 7).

## 4. Discussion

Occupational stress is regarded as particularly harmful to the social psychology of the working population, and a high incidence rate is observed in the global population. A study conducted by Dalia Desouky examined the occupational stress of Egyptian teachers and found that all teachers were suffering from at a mild, or higher, level of occupational stress, and 67.6% of teachers experienced severe occupational stress [37]. Wu Hui found that the rate of occupational stress among train drivers was 72.84% [38]. A survey examining occupational stress among modern service workers in Shanghai, China, showed that 50.1% of workers experience occupational stress [39]. While occupational stress affects physical and mental health, it also adversely impacts upon the working ability and quality of life of the occupational population, placing a huge economic burden on the country and society [40]. 

Copper-nickel miners belong to a special professional group, whose quality of life is strongly related to the safety of production within the mineral industry, which also affects the rapid development of the national economy. The theory of effort–reward imbalance points out that if the worker’s effort is greater than the reward he receives, there will be work pressure. That is to say, when the worker spends a lot of time and energy in his work and the task is difficult but they get less labor remuneration, promotion space, and respect than expected, it is easy to lead to occupational stress. Copper-nickel miners usually work in remote locations, in harsh working environment, with heavy tasks and monotonous and repetitive work tasks. However, their social status is low, promotion space is limited, and work income is relatively low. Relative to their efforts in the work, they get lower returns. Miners in this long-term imbalanced effort–reward state will inevitably experience occupational stress [28]. The survey presented here revealed that 42.65% of copper-nickel miners experience occupational stress, suggesting that occupational stress is prevalent among this particular working population. The higher the level of occupational stress, the lower the quality of life of miners, indicating that occupational stress is a risk factor that can diminish their quality of life.

This survey investigated occupational stress levels among different types of miners employed by copper and nickel mines. The results indicate that the occupational stress level of male miners is higher than that of female miners, which may be due to the fact that female miners in the copper-nickel mine largely include those who are exposed to less occupationally harmful factors, for example those who are employed as administrative staff, and logistics personnel [41]. Copper-nickel miners aged between 30 and 34 years had the highest level of occupational stress. Copper-nickel miners from this age group, which represents the backbone of the industry, entered this type of employment at a mature stage, after having gained production technology and labor experience. At the same time, while considering a change of career, they are eager to seek greater promotion opportunities or to continuously increase their personal income. However, in practice, due to limited promotional opportunities and competitive pressure, their income cannot satisfy their expectations. As a result, the effort does not match the reward, which increases their occupational stress levels [42]. The most serious form of occupational stress was found within the junior college group. Among the copper-nickel miners surveyed, most are skilled workers, and include locomotive operators, grab workers, electricians, and fitters. All of these copper-nickel miners have studied in vocational and technical training schools, and obtained corresponding academic qualifications prior to commencing their employment. Most of the work consists of manual labor, and tasks are relatively repetitive and require shift work. However, monotonous work and the labor organization system are the main sources of tension. Therefore, most copper-nickel miners with a junior college-level education experience serious levels of occupational stress [43].

Comparisons among mining, beneficiation, and smelting units show that occupational stress is highest among copper and nickel miners who work in smelting units, and their quality of life is low. In this investigation, the work of miners in smelting units is more intense and entails more complicated tasks than other units. During the labor production prcoess, such miners are exposed to more occupationally harmful factors (i.e., high temperature, chemical substances, irritant gases, etc.) than in other units. In addition, changes have taken place in the salary distribution assessment system of smelting and other units. Higher levels of tension and a lower quality of life among those within the same labor remuneration system may also be attributed to neglect by management.

This study found that the higher the level of occupational stress among miners, the lower their quality of life. This result suggests that reducing the level of occupational stress among miners can improve their quality of life. Multiple linear regression analysis showed that age, level of education, income, and occupational stress levels affect quality of life among copper-nickel miners. The older the age, the lower the income, the higher the education level, and the higher the degree of occupational stress, the worse the quality of life.

Therefore, relevant departments can be encouraged to foster a healthy organizational system, improve production management, enhance the working environment, strengthen training and psychological interventions for miners, and improve the living and cultural facilities of miners in order to alleviate occupational stress among copper and nickel miners and improve their quality of life.

The present study has some limitations. It is not known if these results can be extrapolated to other regions of China, other countries or other industries. Cross-sectional studies do not provide a precision basis for establishing causality. The relationship between the variables of interest may be influenced by other variables. In the future, further studies with even large sample sizes are needed.

## 5. Conclusions

In conclusion, this study found that copper-nickel miners are generally exposed to occupational stress, and identified gender, age, education, monthly income, and the specific operating unit as influencing factors. In addition, occupational stress influenced the quality of life of all miners. Measures should be taken to alleviate occupational stress among copper and nickel miners in order to improve their quality of life.

## Figures and Tables

**Table 1 ijerph-16-00353-t001:** Characteristics of the copper-nickel worker sample population.

Items	Groups	Case Number	Percentage (%)
Sex	Male	1635	88.0
	Female	222	12.0
Age (years)	<25	489	26.3
	25~	444	23.9
	30~	219	11.8
	35~	195	10.5
	40~	261	14.1
	45~	249	13.4
Education level	Junior high school and below	339	18.3
	High school	435	23.4
	Junior college	885	47.7
	Bachelor’s degree or above	198	10.7
Marital status	Unmarried	672	36.2
	Married	1167	62.8
	Divorced	18	1.0
Income	<2500	348	18.7
(yuan)	2500~	771	41.5
	3000~	225	12.1
	3500~	168	9.0
	4000~	315	17.0
Type of work	Mining unit	441	23.7
	Mineral processing unit	267	14.4
	Smelting unit	546	29.4
	Safety department	57	3.1
	Infrastructure department	111	6.0
	Institutional department	105	5.7
	Logistics department	51	2.7
	Other	279	15.0
Total		1857	100%

**Table 2 ijerph-16-00353-t002:** Comparison of occupational stress levels in different populations. ERI: Effort–Reward Imbalance.

Items	Groups	ERI > 1	ERI ≤ 1	Occupational Stress Detection Rate (%)	Rank Mean Value	Chi-Squared Value	*p*-Value
Sex	Male	716	919	43.79	945.29	7.300	0.004
	Female	76	146	34.23	907.10		
Age (years)	<25	194	295	39.67	901.36	11.472	0.043
	25~	192	252	43.24	934.51		
	30~	115	104	52.51	1020.57		
	35~	84	111	43.08	932.97		
	40~	105	156	40.23	906.53		
	45~	102	147	40.96	913.35		
Education level	Junior high school and below	120	219	35.40	861.6	21.093	<0.001
	High school	182	253	41.84	921.48		
	Junior college	421	464	47.57	974.69		
	Bachelor’s degree or above	69	129	34.85	856.57		
Marital status	Unmarried	272	400	40.48	908.82	2.313	0.315
	Married	511	656	43.79	939.57		
	Divorced	9	9	50.00	997.25		
Income (yuan)	<2500	157	191	45.11	951.89	8.693	0.069
	2500~	350	421	45.40	954.50		
	3000~	96	159	37.65	882.55		
	3500~	69	99	41.07	914.35		
	4000~	120	195	38.10	886.71		

**Table 3 ijerph-16-00353-t003:** Comparison of occupational stress among copper and nickel miners working in mining, ore dressing, and smelting units.

Department	ERI > 1	ERI ≤ 1	Occupational Stress Detection Rate (%)	Rank Mean Value	Chi-Squared Value	*p*-Value
Mining unit	159	282	36.05	598.56	6.043	0.049
Ore dressing unit	114	153	42.70	640.21		
Smelting unit	237	309	43.41	644.66		

**Table 4 ijerph-16-00353-t004:** Comparison of the quality of life of copper and nickel miners working in mining, ore dressing, and smelting units.

Department	PF	RP	BP	GH	VT	SF	RE	MH
Mining unit	732.91	696.84	708.80	718.53	649.09	608.06	660.55	612.01
Ore dressing unit	656.56	617.24	674.83	719.24	741.04	730.09	585.06	716.44
Smelting unit	528.15	576.52	538.70	509.12	554.54	593.03	621.56	596.52
*Z* value	85.014	37.418	61.996	103.684	50.536	29.052	9.821	21.066
*p*-Value	<0.001	<0.001	<0.001	<0.001	<0.001	<0.001	0.007	<0.001

Notes: PF, physiological functioning; RP, role–physical functioning; BP, bodily pain; GH, general health; VT, vitality; SF, social functioning; RE, role–emotional functioning; MH, mental health.

**Table 5 ijerph-16-00353-t005:** Comparison of the quality of life among stressed (an ERI score > 1) and non-stressed (an ERI score ≤1) copper-nickel miners.

Occupational Stress Group	PF	RP	BP	GH	VT	SF	RE	MH
ERI > 1	799.15	810.42	788.75	801.98	755.65	825.28	820.23	798.71
ERI ≤ 1	1025.56	1017.19	1033.30	1023.46	1057.91	1006.13	1009.89	1025.89
*Z*-value	−9.267	−9.537	−9.901	−8.815	−12.070	−7.355	−8.581	−9.061
*p*-Value	<0.001	<0.001	<0.001	<0.001	<0.001	<0.001	<0.001	<0.001

Notes: PF, physiological functioning; RP, role–physical functioning; BP, bodily pain; GH, general health; VT, vitality; SF, social functioning; RE, role–emotional functioning; MH, mental health.

**Table 6 ijerph-16-00353-t006:** Assignment of factor-specific variables.

Variable	Name	Assignment
y	Quality of life	Accurate values
x1	Sex	0 = male, 1 = female
x2	Age	Accurate values
x3	Education level	0 = Junior high school and below, 1 = High school, 2 = Junior college, 3 = bachelor’s degree or above
x4	Income	Accurate values
x5	Over-commitment	Accurate values
x6	ERI	Accurate values

**Table 7 ijerph-16-00353-t007:** The effects of quality of life -related factors among copper-nickel miners according to the results of the multiple linear regression analysis.

Variable	*B*	Beta	*t*	*p*	95% CI	
(constant)	846.802	-	36.750	<0.001	801.909	891.694
Sex	3.170	−0.009	−0.389	0.697	−19.150	12.809
Age	−0.777	−0.061	−2.716	0.007	−1.338	−0.216
Education level	−11.549	−0.087	−3.906	<0.001	−17.348	−5.750
Income	0.014	0.129	5.683	<0.001	0.009	0.019
Over-commitment	−1.501	−0.039	−1.714	0.087	−3.218	0.216
ERI	−196.930	−0.354	−15.712	<0.001	−221.512	−172.349

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
