# Peer review of "The Status of Occupational Stress and Its Influence the Quality of Life of Copper-Nickel Miners in Xinjiang, China"

_ijerph, 2019, doi:10.3390/ijerph16030353_

Round 1
Reviewer 1 Report
This is an interesting study to investigate the status of occupational stress and its influence on the quality of life of copper-nickel miners. It requires some corrections.
First of all, analytical analyses were inconsistent and did not meet the purpose of the study. According to the purpose quality of life was the dependent variable and occupational stress (ERI) was a main independent variable. However, in multiple linear regression (table 7), you treated ERI as dependent variable. This analysis was not enough to draw conclusion.
Additional explanations are needed for the originality of the study.
In Materials and Methods, “validity rate”(line 102) should be replaced by response rate.
You had better present the reference of occupational stress measurement. There are a couple of versions of ERI model questionnaire.
I don’t know “Intrinsic effort (I, six items)”. Was it ‘overcommitment’ in ERI model? Can it be a dependent variable in the multiple linear regression (table 7)?
In table 7, What does ** mean?
Author Response
Thank you very much for your valuable comments on the manuscript. We have revised the manuscript accordingly according to your comments.
(a)We modified the multiple linear regression part and analyzed the quality of life as dependent variable and ERI as independent variable.
(b)We have supplemented the content of originality.
(c)We have replaced "validity rate" with "response rate."
(d)We used the improved Chinese version of the Effort-Reward Imbalance Questionnaire, which has been tested in a number of people with good reliability and validity. we have supplemented the references.
(e)We believe that Over-commitment is also a factor affecting quality of life, so we regard it as an independent variable of multiple linear regression analysis. There are also similar views in the references.
Chu Kequn, Song Guoping. Transfer, Expansion and Prospect of Work Stress Theory of Pay-Feedback Imbalance [J]. Progress in Psychological Science, 2016,24(2): 242-249.DOI: 10.3724/SP.J.1042.2016. 00242.(In Chinese)
(f)The previous "**" may have been an error, and we have deleted it.
Reviewer 2 Report
The study is very interesting because provide data about occupational stress on a population (miners) rarely investigated. However, we believe the methodology of the research must be improved. Below are some comments for the authors.
Basing on the review of the previous research on the field, it would be expected that some hypothesis for the study were proposed, but only a general objective is formulated to study the relation between occupational stress and quality of life. Even only objectives were formulated, it would be convenient they were more specific and accurate in order to guide and anticipate what is done later in the results section.
In participants section it is identified the city and the region where the study was done, but not the country. The country is not mentioned until the end of the manuscript.
It is specified the number of participants but not so clearly the number of mines where the data was gathered. In participants section seems as all participants come from the same mine (copper-nickel mine in Hami City), but below is said “we contacted the organizations that were selected to participate in this study in order to foster cooperation and facilitate a small-scale pre-investigation”.
The probability P is better to write it in lowercase (p).
It would be also convenient to be more precise and accurate with the values of p. Not to say p < .05 or p <.01 but to write the exact value of p. Generally, only such a general expression is accepted when the value of p is less than .001.
In line 113 the name of the author of the scale must be followed by the reference in parenthesis.
The main objective of the research is to study the effect of the occupational stress on the quality of life of the miners, however in the end of the result section is unexpectedly investigated the effect of the quality of life on occupational stress (Table 6).
In linear multiple regression predictors are assumed to be quantitative variables. Marital status is a categorical variable with three not ordered values. In addition, sex and educational level are reduced to two values (1 and 2). It would be better to transform the original values into 0 and 1 to facilitate the interpretation of the estimates. In any case, logistic regression is a more appropriate statistical model for this kind of data.
In Table 1 it is observed that the miners distribute along seven types of work. However only three of them are selected to compare. It is not clear the reason to exclude some types of work from the statistical analysis.
The rank sum test would better recognized if its authors were identified, probably Mann-Whitney.
In Tables 4 and 5 it would be appreciated a footnote explaining what is PF, RP, BP, etc.
Author Response
Thank you very much for your valuable comments on the manuscript. We have revised the manuscript accordingly according to your comments.
(a) We have made some additions to the objectives, and we have also made some modifications to the title and content of the manuscript.
(b) The country in which the survey was conducted was supplemented.
(c) Data from the survey were collected at a Copper-Nickel Mine in Hami, Xinjiang Uygur Autonomous Region, China. Before the investigation, we got in touch with the units under investigation in order to get their cooperation and ensure the smooth development of the investigation. We have revised the corresponding content.
(d) We have changed "P" to "p".
(e) We have accurately calculated the p value in the manuscript as required.
(f) We have supplemented references to the Effort-Reward Imbalance Questionnaire.
(g) We modified the multiple linear regression part and analyzed the quality of life as dependent variable and ERI as independent variable.
(h) We have revised the relevant content according to the expert's opinions. Because the quality of life is a measurement data in this survey, it cannot be classified in the questionnaire version, so we use multiple linear regression.
(i)There are 7 types of work in this survey, but the time and dose of exposure to occupational hazards in Safety department, infrastructure department, institutional department and logistics department are much lower than that in mining unit, mining unit and smelting unit, so only three main types of work are selected for comparison and analysis.
(j) We have modified the rank sum test.
(k) We added footnote under Tables 4 and 5.
Reviewer 3 Report
The design and methods of research do not let to establish causality (the authors of the publication agree with this conclusion in line 257). The title of the publication should be changed, the content of the article should be edited.
The statement that "the quality of life affect the occupational stress" (189 - 190 line) is incorrect and does not fit the theory introduced in the Introduction.
It seems that the authors did the mistake in regression method (please, look to the dependent and independent variables list).
The authors writes P talking about the significance level in the article (159; 160 line and etc.), but usually we write p.
Discussion of the results could be deeper.
Author Response
Thank you very much for your valuable comments on the manuscript. We have revised the manuscript accordingly according to your comments.
(a) We have made some modifications to the title and content of the manuscript. This study explores the impact of occupational stress on quality of life. Although cross-sectional studies cannot get accurate causal relationship, they can provide some clues and basis for it.
(b) We have revised the corresponding content of "189-190 line".
(c) We modified the multiple linear regression part and analyzed the quality of life as dependent variable and ERI as independent variable.
(d) We have supplemented the relevant content.
Round 2
Reviewer 1 Report
Many parts of manuscript have been modified according to review. You can add some discussion from the ERI model perspective.
Author Response
Thank you very much for your valuable comments on this paper. We have supplemented the relevant contents according to the expert' suggestion.
Reviewer 2 Report
In line 119, "ming unit" is repeated twice. We think it is missing "ore dressing".
In lines 199-204 the names of the variables must correpond with the names used in Tables 6 and 7.
In multiple linear regression only continuous variables can be used as predictors. Exceptionally categorical variable with only two values can be used as well. Clearly maritual status is not a continous variable nor a categorical variable with only two values. It has thre values: unmarried, married and divorced. Therefore it must be eliminated of the analysis. Please, note that it has not effect on quality of life. To estimate the effect of marital status it is better to use analysis of variance.
Author Response
Thank you very much for your valuable comments on the manuscript. We have revised the manuscript accordingly according to your comments.
(a)We have changed "mining unit" to "ore dressing unit ".
(b) We have revised the corresponding content of " 199-204 line".
(c) We revised the regression part according to the expert'opinion.